# Beta-blocker and survival in patients with lung cancer: A meta-analysis

**Zhen Lei[1], Weiyi Yang[1], Ying Zuo[ID][2]***

**1** Department of Respiratory and Critical Care Medicine, the Affiliated Hospital of North Sichuan Medical College, Nanchong, Sichuan, China, **2** Department of Endocrinology and Metabolism, the Affiliated Hospital of North Sichuan Medical College, Nanchong, Sichuan, China

* zuoying_cb89@tom.com

## Abstract

### Background

Beta-blocker (BB) is suggested to have anticancer efficacy. However, the potential influence of BB use on overall survival (OS) in patients with lung cancer remains undetermined. We aimed to evaluate the above relationship in an updated meta-analysis.

### Methods

Observational studies comparing OS between users and non-users of BB with lung cancer were identified by search of PubMed, Embase, and Cochrane's Library. A random-effect model was used to pool the results.

### Results

Ten retrospective cohort studies with 30870 patients were included. Overall, BB use was not associated with significantly improved OS in lung cancer (hazard ratio [HR] = 1.02, 95% confidence interval [CI]: 0.98 to 1.06, p = 0.33) with moderate heterogeneity ($I^2$ = 29%). Stratified analyses showed similar results in patients with non-small cell lung cancer and small cell lung cancer, in studies with BB use before and after the diagnosis of lung cancer, and in studies with or without adjustment of smoking. Use of BB was associated with improved OS in patients with stage III lung cancer (HR = 0.91, 95% CI: 0.85 to 0.98, p = 0.02) and in patients that did not receive surgery resection (HR = 0.78, 95% CI: 0.64 to 0.96, p = 0.02), while use of non-selective BB was associated with worse OS (HR = 1.14, 95% CI: 1.01 to 1.28, p = 0.03).

### Conclusions

This meta-analysis of retrospective cohort studies does not support a significant association between BB use and improved OS in lung cancer.

**Data Availability Statement:** All relevant data are within the manuscript and its Supporting Information files.

**Funding:** The author(s) received no specific funding for this work.

**Competing interests:** The authors have declared that no competing interests exist.

## Introduction

Lung cancer is one of the most common cancers worldwide [1, 2]. Moreover, lung cancer is also the leading cause of cancer-specific mortality for the global population, which results in more than 1.7 million deaths annually [1, 2]. Lung cancer can be broadly divided into two categories, including small-cell lung cancer (SCLC) which accounts for about 15% of the cases, and non-small cell lung cancer (NSCLC) which accounts for about 85% of the cases [3]. Although current comprehensive treatment strategies, including surgery resection, chemotherapy, radiotherapy, immune therapy, and targeted therapy have shown promising efficacy in patients of lung cancer, the prognosis of these patients remains poor [4, 5]. Therefore, identification of additional treatments for lung cancer remains of great clinical significance [6, 7]. Preclinical studies demonstrated that chronic stress accelerates the process of carcinogenesis via increasing the serum level of catecholamine and subsequent activation of adrenergic receptors in target tissues, including cancer cells [8]. Indeed, the activation of beta adrenergic receptor axis in cancer has been related with stimulated angiogenesis, enhanced expression of genes in metastasis and inflammation, upregulated cancer cell proliferation, and modulated tumor immune microenvironment, which are all key pathophysiological processes in tumorigenesis, angiogenesis, and tumor metastasis [8–10]. Accordingly, use of beta-blocker (BB) has been hypothesized to confer anticancer efficacy by targeted inhibition of beta adrenergic receptor axis in tumor development and progression [11]. Previous studies have confirmed that use of BB is related with reduced incidence of liver cancer [12] and improved survival in prostate cancer [13]. However, studies evaluating the association between BB use and survival of lung cancer showed inconsistent results [14–23]. An early study showed that the incidental use of BB among patients with NSCLC treated with definitive radiation therapy was associated with improved survival [17]. However, other studies failed to detect a potential benefit of BB use on survival in patients with lung cancer [14–16, 18–23]. Moreover, although a few meta-analyses have been published to evaluate the influence of BB use on survival of lung cancer [24–27], these meta-analyses only included 3–7 studies published before 2018. Some recently published studies have not been included [22, 23]. Moreover, the potential influences of study characteristics on this association, such as category and stage of lung cancer, timing of BB use, category of BB, and smoking status of the patients, were rarely investigated. Therefore, this updated meta-analysis was performed to provide a systematic evaluation of the association between BB use and survival of lung cancer and explore the potential influences of above study characteristics on the outcome.

## Methods

The design, implementation, and report of the meta-analysis was in accordance with the MOOSE (Meta-analysis of Observational Studies in Epidemiology) [28] and Cochrane's Handbook [29] guidelines.

### Literature search

Electronic databases of PubMed, Embase, and the Cochrane's Library were searched using the combination of the following terms: (1) "adrenergic beta antagonist" OR "beta blockers" OR "beta antagonist" OR "beta adrenoreceptor antagonist" OR "beta adrenergic receptor antagonist" OR "beta adrenergic blocking agent" OR "adrenergic beta-1 receptor antagonists" OR "acebutolol" OR "alprenolol" OR "atenolol" OR "betaxolol" OR "bisoprolol" OR "bunolol" OR "bupranolol" OR "Bucindolol" OR "carteolol" OR "celiprolol" OR "Carvedilol" OR "dihydroalprenolol" OR "esmolol" OR "iodocyanopindolol" OR "labetalol" OR "levobunolol" OR "metipranolol" OR "metoprolol" OR "nadolol" OR "Nebivolol" OR "oxprenolol" OR "penbutolol" OR

"practolol" OR "pindolol" OR "propranolol" OR "sotalol" OR "timolol"; (2) "lung cancer"; and (3) "survival" OR "prognosis" OR "mortality" OR "recurrence" OR "recurrent" OR "death" OR "metastasis" OR "progression" OR "hazard ratio" OR "surgery" OR "operation" OR "risk". We expanded the search terms regarding the outcomes of the patients to avoid missing of potential related studies. The search was limited to human studies published in English or Chinese. The reference lists of original and review articles were also analyzed manually. The final literature search was performed on January 15, 2020.

## Study selection

Studies were included if they met the following criteria: (1) published as full-length article; (2) designed as studies with longitudinal follow-up, such as randomized controlled trials (RCTs), cohort studies, nested case-control studies, and post-hoc analyses of RCTs, with a minimal follow-up duration of one year; (3) included patients with lung cancer; (4) compared the survival outcome between users and non-users of BB with lung cancer; (5) documented the incidence of overall survival (OS) during follow-up; and (6) reported the adjusted hazard ratios (HRs, at least adjusted for age and gender) and their corresponding 95% confidence intervals (CIs) for the above outcomes in users and non-users of BB with lung cancer. Reviews, editorials, preclinical studies, cross-sectional studies, conference abstracts were excluded.

## Data extracting and quality evaluation

Literature search, data extraction, and study quality assessment were independently performed by two authors according to the predefined inclusion criteria. If inconsistency occurred, discussion with the corresponding author was suggested to resolve these issues. The following data were extracted: (1) name of the first author, publication year, study location, and study design; (2) characteristics and number of patients with lung cancer, mean age, proportion of male, cancer stage, definition of BB use, and follow-up duration; and (3) outcomes reported, and variables adjusted when presenting the HRs. The quality of observational study was evaluated using the Newcastle-Ottawa Scale (NOS) [30]. This scale ranges from 1 to 9 stars and judges the quality of each study regarding three aspects: selection of the study groups; the comparability of the groups; and the ascertainment of the outcome of interest.

## Statistical analyses

The association between BB use and survival outcomes in lung cancer patients was measured by adjusted HRs. To stabilize its variance and normalized the distribution, HR data and its corresponding stand error (SE) from each study was logarithmically transformed [29]. The Cochrane's Q test was performed to evaluate the heterogeneity among the include cohort studies [29, 31], and the $I^2$ statistic was also calculated. A significant heterogeneity was considered if $I^2 > 50\%$. A random effect model was used to pool the results since this model has been indicated to incorporate the potential heterogeneity of the included studies and therefore could provide a more generalized result. Sensitivity analysis by omitting one study at a time was performed to evaluate the stability of the results [29]. Predefined stratified analyses were performed to evaluate the potential influences of study characteristics on the outcomes, including categories of lung cancer, tumor stage, with and without of surgical resection, timing of BB use, categories of BB (selective [SBB] or non-selective BB [NSBB]), and with or without of adjustment of smoking status when presenting the results [32]. We defined BB use as before or after diagnosis of lung cancer according to the time when the patients started using of the medication. Potential publication bias was assessed by visual inspection of the symmetry of the

funnel plots, complemented with the Egger regression test [33]. The RevMan (Version 5.1; Cochrane Collaboration, Oxford, UK) and STATA software were used for the statistics.

## Results

### Literature search

The flowchart of database search was shown in **Fig 1**. Briefly, 922 studies were obtained from database search, and 898 of them were excluded primarily because they were irrelevance to the

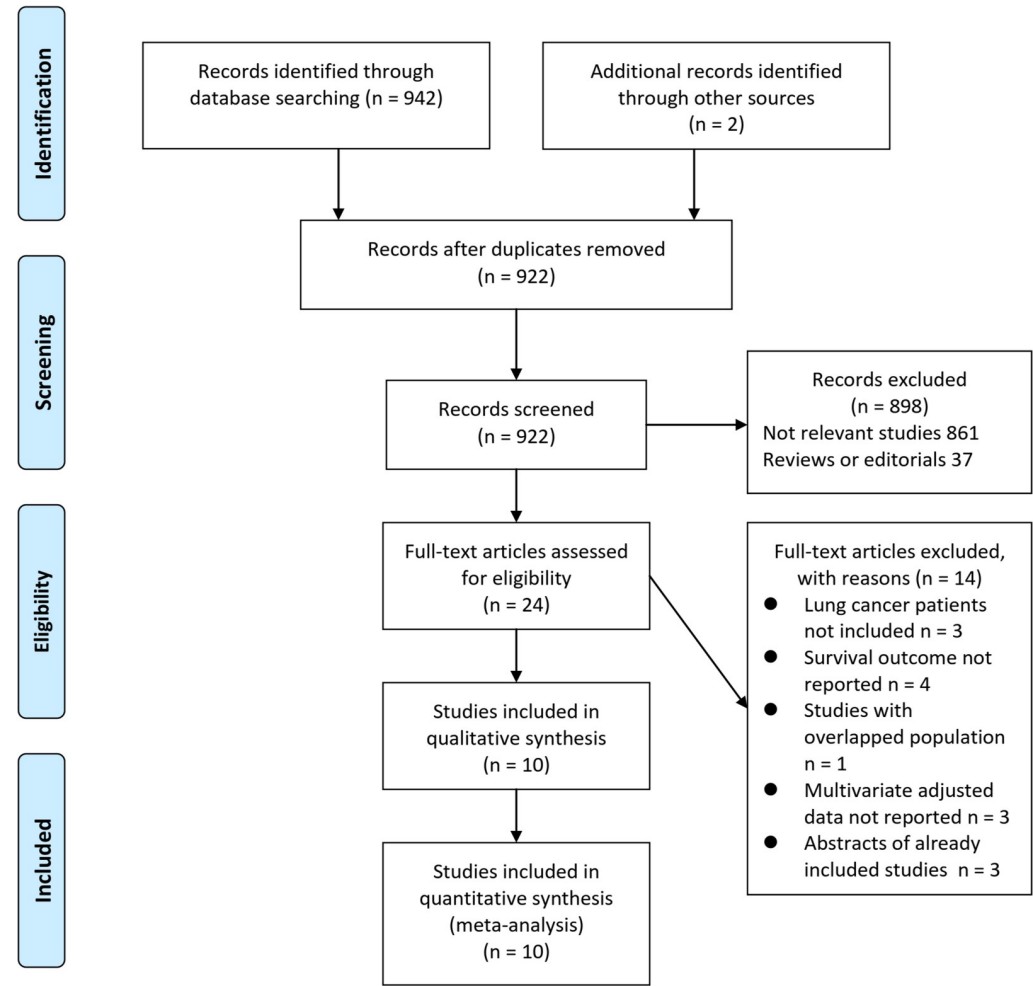

PRISMA FLOW DIAGRAM

From: Moher D, Liberati A, Tetzlaff J, Altman DG, The PRISMA Group (2009). *Preferred Reporting Items for Systematic Reviews and Meta-Analyses: The PRISMA Statement. PLoS Med 6(7): e1000097. doi:10.1371/journal.pmed1000097*

For more information, visit www.prisma-statement.org.

**Fig 1. Flowchart of database search and study inclusion.**

aim of the study. For the remaining 24 potential relevant studies that underwent full-text review, 14 were further excluded because three of them did not include lung cancer patients, four did not report survival outcome, one included overlapped patients of other studies, three did not report multivariate adjusted data, and the other three were abstracts of already included studies. Finally, ten studies were included [14–23].

## Study characteristics and quality

Overall, this meta-analysis included ten retrospective cohort studies [14–23] with 30870 patients with lung cancer. No RCT was identified. The characteristics of the included cohorts were shown in **Table 1**. These studies were published between 2011 and 2020, and performed in Europe, North America, and Asia. Five studies included patients with NSCLC [15, 17, 18, 21, 23], while the remaining five included both NSCLC and SCLC [14, 16, 19, 20, 22]. The mean ages of the included patients varied between 56 and 74 years, with proportions of male varying from 50 to 95%. Users of BB were defined as BB use prior or after the diagnosis of lung cancer. The mean follow-up durations varied from 1.6 to 6.5 years. Potential confounding factors, including age, cancer stage at diagnosis, treatments, comorbidities and concurrent medications were adjusted to a varying degree in the included studies. The qualities of the included follow-up studies were generally good, with the NOS ranging from seven to nine points (**Table 2**).

## Overall meta-analysis

Pooling the results of ten included retrospective cohort studies showed that BB use was not associated with significantly affected OS in lung cancer (adjusted HR = 1.02, 95% CI: 0.98 to 1.06, p = 0.33; **Fig 2**) with moderate heterogeneity ($I^2$ = 29%). Sensitivity analysis by omitting one study at a time showed similar results (adjusted HR: 1.01~1.02, p all > 0.05).

## Stratified analyses

Results of stratified analyses showed that BB use was not associated with significantly affected OS in patients with NSCLC (adjusted HR = 0.97, p = 0.55), SCLC (adjusted HR = 1.02, p = 0.77), or mixed categories (adjusted HR = 1.03, p = 0.09; **Fig 3**), in patients with stage I-II (adjusted HR = 0.98, p = 0.73) or stage IV lung cancer (adjusted HR = 1.07, p = 0.43; **Fig 4**), in patients who received surgical resection (adjusted HR = 1.08, p = 0.38; **Fig 5**), in patients using BB prior (adjusted HR = 1.02, p = 0.13) or after the diagnosis of lung cancer (adjusted HR = 0.96, p = 0.66; **Fig 6**), in patients using SBB (adjusted HR = 1.02, p = 0.51; **Fig 7**), and in studies with (adjusted HR = 1.07, p = 0.21) or without the adjustment of smoking status (adjusted HR = 1.01, p = 0.59; **Fig 8**). In addition, BB use was associated with improved OS in stage III lung cancer (HR = 0.91, 95% CI: 0.85 to 0.98, p = 0.02; **Fig 4**) and in patients who did not receive surgery (HR = 0.78, 95% CI: 0.64 to 0.96, p = 0.02; **Fig 5**), while use of non-selective BB seemed to be associated with worse OS in lung cancer patients (HR = 1.14, 95% CI: 1.01 to 1.28, p = 0.03; **Fig 7**).

## Publication bias

The funnel plots for the association between BB and OS of lung cancer were shown in **Fig 9**. The plots were symmetrical on visual inspection, suggesting low risks of publication bias. Results of Egger's regression tests also showed similar results (p = 0.627).

**Table 1. Characteristics of the included patients.**

| Study | Country | Design | Patient characteristics | Sample size | Mean age | Male | Stages | Definition of BB use | Follow-up duration | Outcomes reported | Outcome validation | Variables adjusted | NOS |
|---|---|---|---|---|---|---|---|---|---|---|---|---|---|
| | | | | | years | % | | | years | | | | |
| Shah 2011 | UK | RC | Patients with lung cancer | 436 | NR | NR | I-IV | BB use within 1y prior to the diagnosis of lung cancer | 4.8 | OS | Medical records | Age, gender, smoking status, concurrent medications, and national region | 7 |
| Aydiner 2013 | Turkey | RC | Patients with metastatic NSCLC | 107 | 61.0 | 94.4 | IV | BB use after the diagnosis of lung cancer | 1.5 | OS | Medical records | Age, gender, performance status, histologic subtype, smoking status, comorbidities, and concurrent medications | 7 |
| Wang 2013 | the US | RC | Patients with NSCLC Treated with definitive radiation therapy | 722 | 65.0 | 55.0 | I-III | BB use after the diagnosis of lung cancer | 3.7 | DMFS, DFS, and OS | Medical records | Age, gender, KPS, clinical stage, tumor histology, use of concurrent chemotherapy, radiation dose, gross tumor volume, comorbidities, and aspirin | 9 |
| Holmes 2013 | Canada | RC | Patients with lung cancer | 4241 | 71.0 | 52.3 | I-IV | BB use within 1y prior to the diagnosis of lung cancer | 4.2 | OS | Medical records | Age, gender, tumor stage, history of cancer, and area of residence | 7 |
| Cata 2014 | the US | RC | Patients with NSCLC who underwent resection | 435 | NR | 52.4 | I-IIIa | BB use after the diagnosis of lung cancer | 4.4 | DFS, and OS | Medical records | Age, gender, tumor stage, BMI, smoking status, preoperative and postoperative radiation, thoracotomy, perioperative blood transfusions, and comorbidities | 8 |
| Springate 2015 | UK | RC | Patients with lung cancer | 1326 | NR | 55.3 | I-IV | BB use within 1y prior to the diagnosis of lung cancer | 3.2 | OS | Medical records | Age, gender, smoking status, concurrent medications, and national region | 7 |
| Yang 2017 | China | RC | Patients undergoing chemoradiotherapy with inoperable stage III NSCLC | 606 | 56.7 | 88.6 | III | BB use after the diagnosis of lung cancer | 1.8 | OS | Medical records | Age, gender, smoking status, alcohol intake, tumor stage, tumor pathology, and chemotherapy | 7 |

(*Continued*)

**Table 1.** (Continued)

| Study | Country | Design | Patient characteristics | Sample size | Mean age | Male | Stages | Definition of BB use | Follow-up duration | Outcomes reported | Outcome validation | Variables adjusted | NOS |
|---|---|---|---|---|---|---|---|---|---|---|---|---|---|
| | | | | | years | % | | | years | | | | |
| Weberpals 2017 | Germany | RC | Patients with lung cancer | 2500 | 70.4 | 71.9 | I-IV | BB use within 1y prior to or after the diagnosis of lung cancer | 6.5 | OS | Medical records | Age, gender, year of diagnosis, socio-economic status, comorbidities, cancer treatment, best supportive care, cancer stage, histology, previous cancer, concurrent medications | 9 |
| Musselma 2018 | Canada | RC | Patients with lung cancer who underwent resection | 2068 | 73.7 | 54.5 | I-III | BB use within 1y prior to the diagnosis of lung cancer | 3.6 | OS | Medical records | Age, gender, socioeconomic status and CCI | 6 |
| Udumyan 2020 | Sweden | RC | Patients with NSCLC | 18429 | 69.1 | 50.8 | I-IV | BB use within 1y prior to the diagnosis of lung cancer | 1.6 | OS | Medical records | Age, gender, tumor history, tumor stage, education, marital status, region of residence, comorbidities and concurrent medications | 9 |

BB, beta-blocker; NOS, Newcastle-Ottawa Scale; UK, United Kingdom; US, United States; RC, retrospective cohort; NSCLC, non-small cell lung cancer; NR, not reported; OS, overall survival; DFS, disease-free survival; DMFS, distant metastasis-free survival; KPS, Karnofsky Performance Score; CCI, Charlson Comorbidity Index; BMI, body mass index.

## Discussion

In this meta-analysis of observational studies, we found that BB use was not significantly associated with improved OS in patients with lung cancer. Subsequently, results of comprehensive stratified analyses showed consistent results in patients with NSCLC or SCLC, in studies with BB use before or after the diagnosis of lung cancer, and in studies with or without adjustment of smoking. Limited evidence from subgroups of 2~4 studies showed that use of BB might be associated with improved OS in patients with stage III lung cancer and in patients without surgery resection, while use of non-selective BB might be associated with worse OS. Taken together, although effects of BB in certain subgroups may be significant, current evidence from retrospective cohort studies did not show a significant association between BB use and improved OS in patients with lung cancer. However, in view of the promising results in pre-clinical studies and the potential limitations of retrospective cohort studies, large-scale prospective cohort studies or even RCTs are needed to validate the potential influence of BB use on survival in patients with lung cancer.

Although four meta-analyses have been published previously regarding the association between BB use and survival in patients with lung cancer, these meta-analyses were based on only 3–7 studies published before 2018, mostly focusing on patients with various malignancies

**Table 2. Details of quality evaluation for the included studies via the Newcastle-Ottawa Scale.**

| Study | Representativeness of the exposed cohort | Selection of the non-exposed cohort | Ascertainment of exposure | Outcome not present at baseline | Control for age and gender | Control for other confounding factors | Assessment of outcome | Enough long follow-up duration | Adequacy of follow-up of cohorts | Total |
|---|---|---|---|---|---|---|---|---|---|---|
| Shah 2011 | 1 | 1 | 1 | 1 | 1 | 0 | 1 | 1 | 1 | 8 |
| Aydiner 2013 | 0 | 1 | 1 | 1 | 1 | 1 | 1 | 1 | 1 | 8 |
| Wang 2013 | 1 | 1 | 1 | 1 | 1 | 1 | 1 | 1 | 1 | 9 |
| Holmes 2013 | 1 | 1 | 1 | 1 | 1 | 0 | 1 | 1 | 1 | 8 |
| Cata 2014 | 0 | 1 | 1 | 1 | 1 | 1 | 1 | 1 | 1 | 8 |
| Springate 2015 | 1 | 1 | 1 | 1 | 1 | 0 | 1 | 1 | 1 | 8 |
| Yang 2017 | 0 | 1 | 1 | 1 | 1 | 1 | 1 | 1 | 1 | 8 |
| Weberpals 2017 | 1 | 1 | 1 | 1 | 1 | 1 | 1 | 1 | 1 | 9 |
| Musselma 2018 | 0 | 1 | 1 | 1 | 1 | 0 | 1 | 1 | 1 | 7 |
| Udumyan 2020 | 1 | 1 | 1 | 1 | 1 | 1 | 1 | 1 | 1 | 9 |

rather than lung cancer patients only [24–27]. Overall, results of this updated meta-analysis were consistent with findings of the previous meta-analyses, which did not show that BB use was significantly associated with improved OS in patients with lung cancer. However, our study has the following strengths as compared with previous meta-analyses. Firstly, unlike most of the previous meta-analyses which focused on the influence of BB use on survival in patients with various cancers [24–26, 34], we included updated observational studies evaluating the association between BB use and OS only in lung cancer patients. A total of 10 studies with 30870 patients were included, which accounts for a much larger sample size than previous meta-analyses. Secondly, only studies with multivariate analyses were included in this meta-analysis. Accordingly, unlike previous meta-analyses which also included studies with univariate analyses [24–26, 34], the finding of this meta-analysis was minimally affected by potential confounding factors as seen in studies based on univariate analyses. Thirdly, the stability of the finding of the meta-analysis was approved by the results of sensitivity analyses, which showed that the overall finding of the meta-analysis was not affected by either of the included studies.

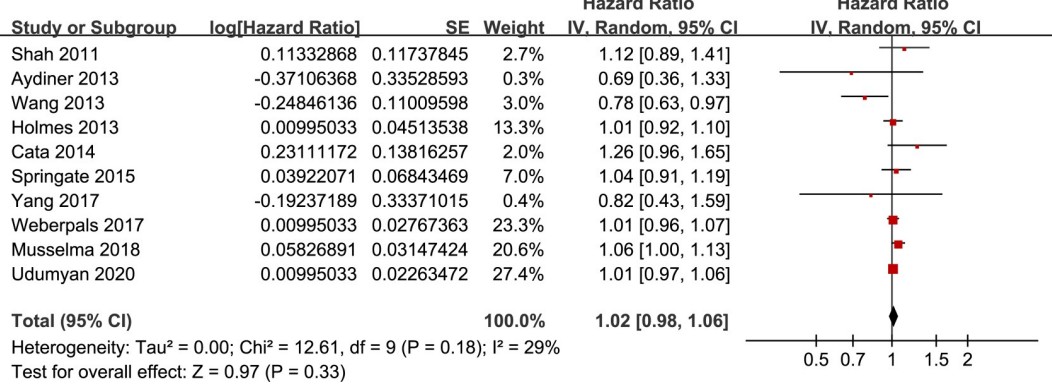

**Fig 2. Funnel plots for the meta-analysis of overall association between BB use and OS of lung cancer.**

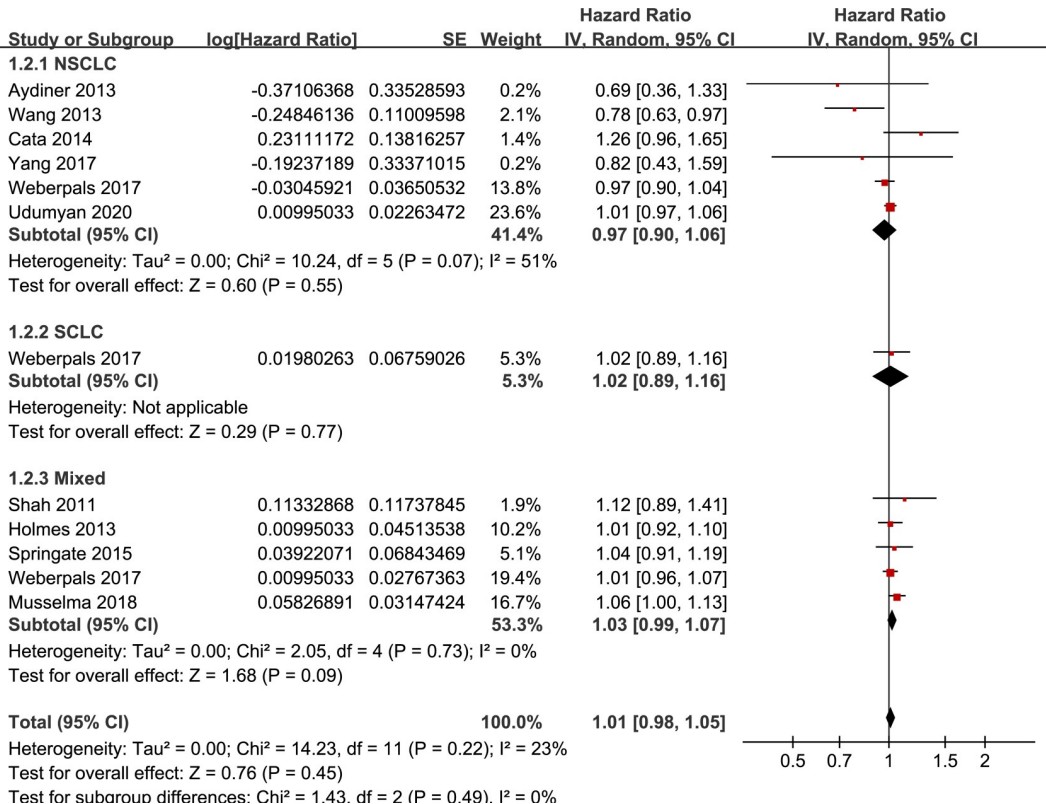

**Fig 3. Stratified analyses for the meta-analysis of the association between BB use and OS of lung cancer according to the category of lung cancer.**

Fourthly, the relatively large number of included studies in the current meta-analysis compared to the previous ones enabled us to perform comprehensive subgroup analyses, which were rarely performed in previous meta-analyses [24–27, 34]. Results of subgroup analyses showed that BB use was not associated with significantly affected OS of lung cancer in patients with NSCLC or SCLC, in studies with BB use before or after the diagnosis of lung cancer, and in studies with or without adjustment of smoking, which were consistent with the results of main analysis. Moreover, subgroup analyses also showed that use of BB might be associated with improved OS in patients with stage III lung cancer and in patients without surgery resection, while use of non-selective BB might be associated with worse OS. These findings may have implications for design of subsequent prospective cohort studies or RCTs, and studies focusing on these subgroups may be considered.

Preclinical studies generally showed that activation of beta adrenergic receptor axis, particularly via beta2 adrenergic receptor is implicated in the pathogenesis and progression of lung cancer via effects on apoptosis, proliferation, angiogenesis, and migration [8]. However, these early studies were mostly performed in pulmonary adenocarcinoma cell lines [35], and subsequent analyses showed that the expression of beta adrenergic receptor may be differently expressed in different histological type of lung cancer, and the response of these cancer cells to the beta adrenergic signal may be different [36]. Lung cancer is a histopathologically heterogeneous disease. However, most of the previous studies categorized lung cancer broadly into SCLC and NSCLC rather than into specific histological type, which may explain the inconsistent results of the included studies. Our subgroup analyses which showed benefits of BB in certain subgroups (stage III lung cancer and those without surgery resection) may also be related

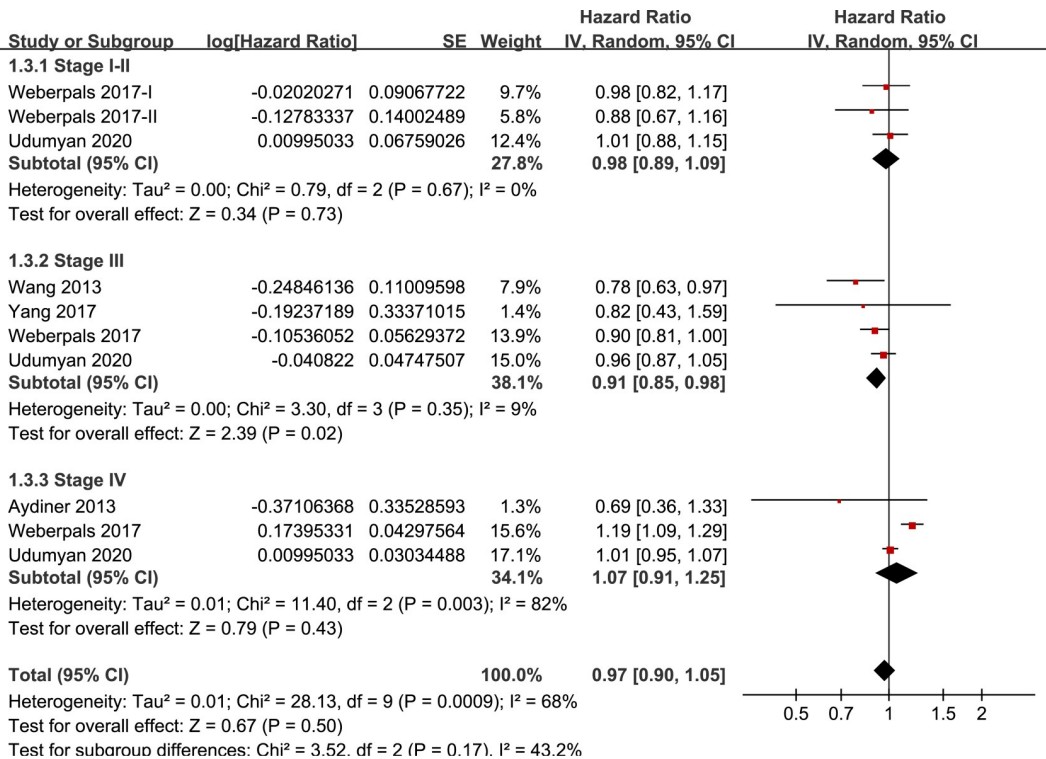

Fig 4. Stratified analyses for the meta-analysis of the association between BB use and OS of lung cancer according to the stage of the tumor.

to the heterogeneous pathological type included in these subgroups. Since surgical resection is not generally considered as standard of care for later stage patients with lung cancer, results of these subgroup analyses may reflect the potential adverse interaction of BB with treatment strategies other than surgery in these patients. Indeed, an early study including 107 patients

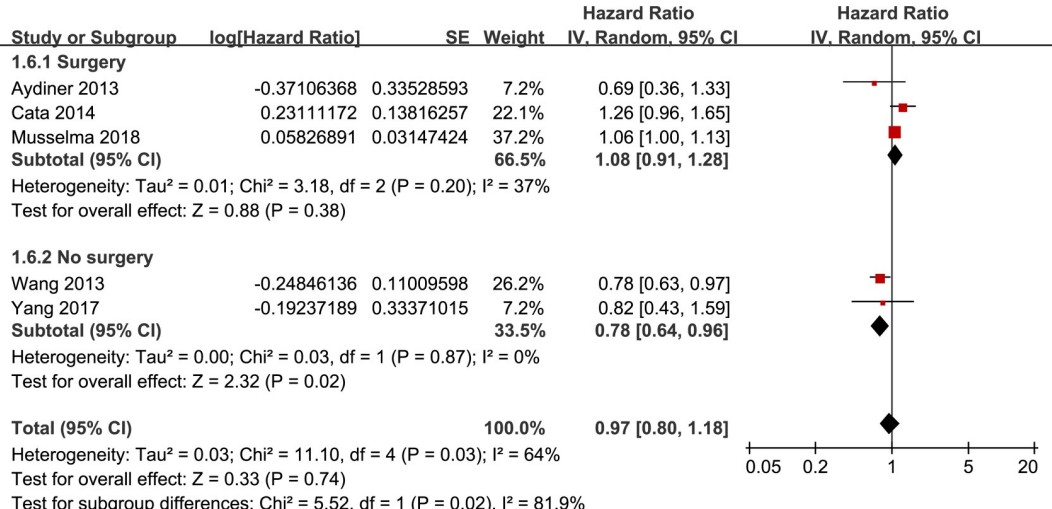

Fig 5. Stratified analyses for the meta-analysis of the association between BB use and OS of lung cancer according to whether surgical resection was performed.

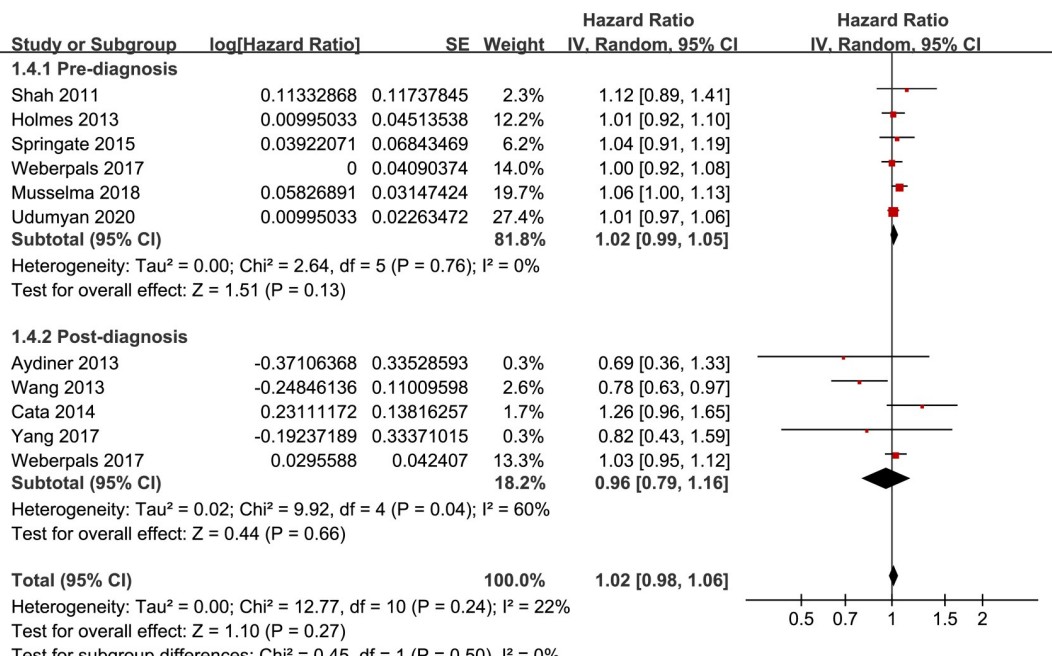

**Fig 6. Stratified analyses for the meta-analysis of the association between BB use and OS of lung cancer according to the timing of BB use.**

with metastatic NSCLC receiving chemotherapy showed that BB use was associated with improved progression-free survival [15]. In addition, a recent retrospective study also showed that BB use was associated with improved prognosis in NSCLC patients that received immune checkpoint inhibitors [37]. Moreover, we found that use of non-selective BB might be associated with worse OS, which seems to be paradoxical to the previous finding from preclinical

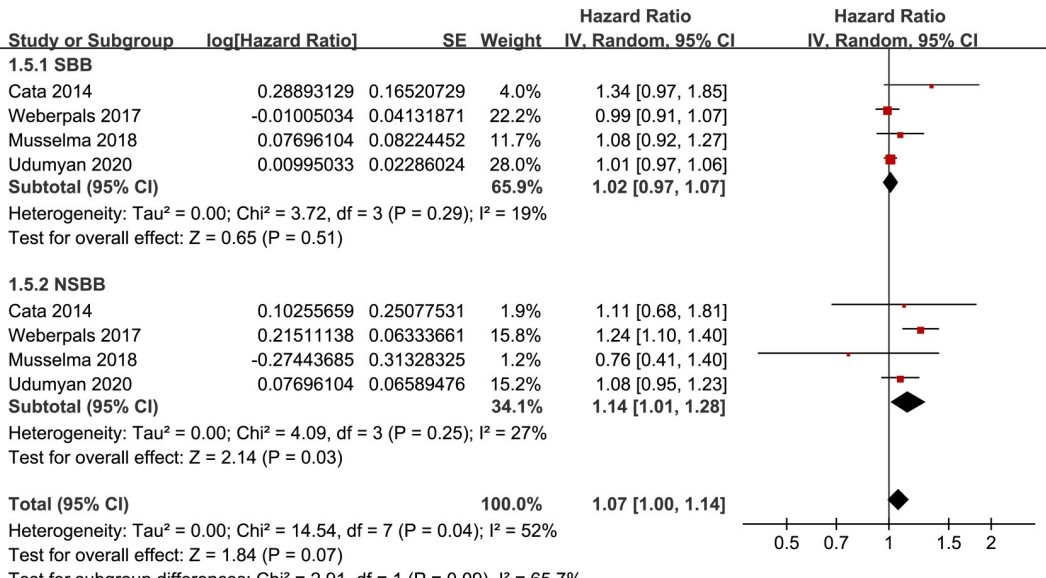

**Fig 7. Stratified analyses for the meta-analysis of the association between BB use and OS of lung cancer according to the category of BB.**

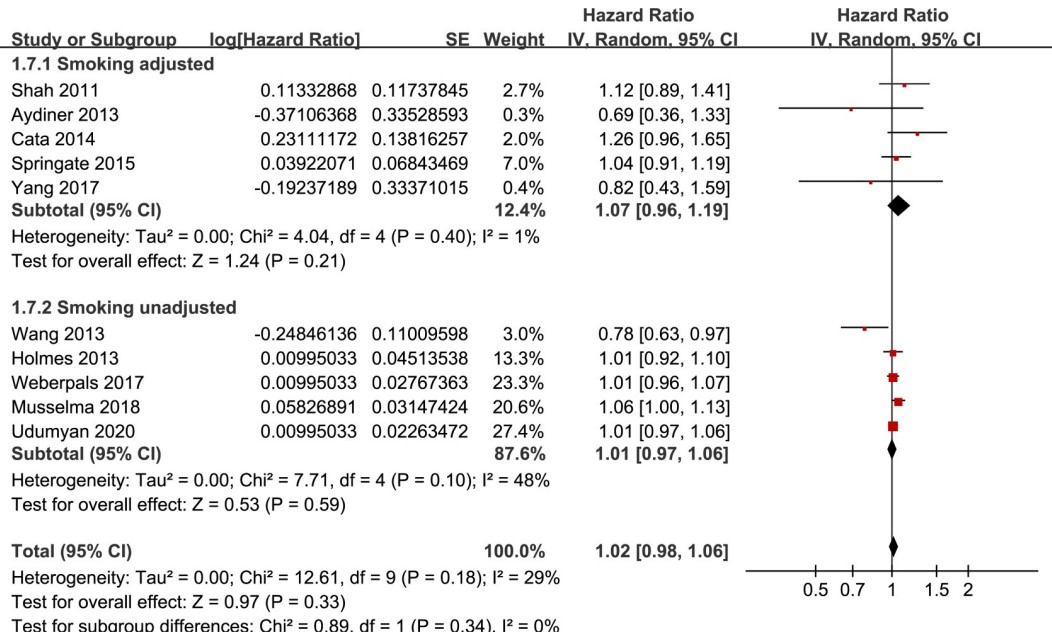

**Fig 8. Stratified analyses for the meta-analysis of the association between BB use and OS of lung cancer according to whether smoking status was adjusted.**

studies that beta2 adrenergic receptor signal is the main pathway involved in cancer progression. The potential explanation for this paradox remains unknown at current stage. However, a previous pooled analysis with individual patient data showed use of propranolol or NSBB was not associated with improved survival in breast cancer [38], which is also not in consistent

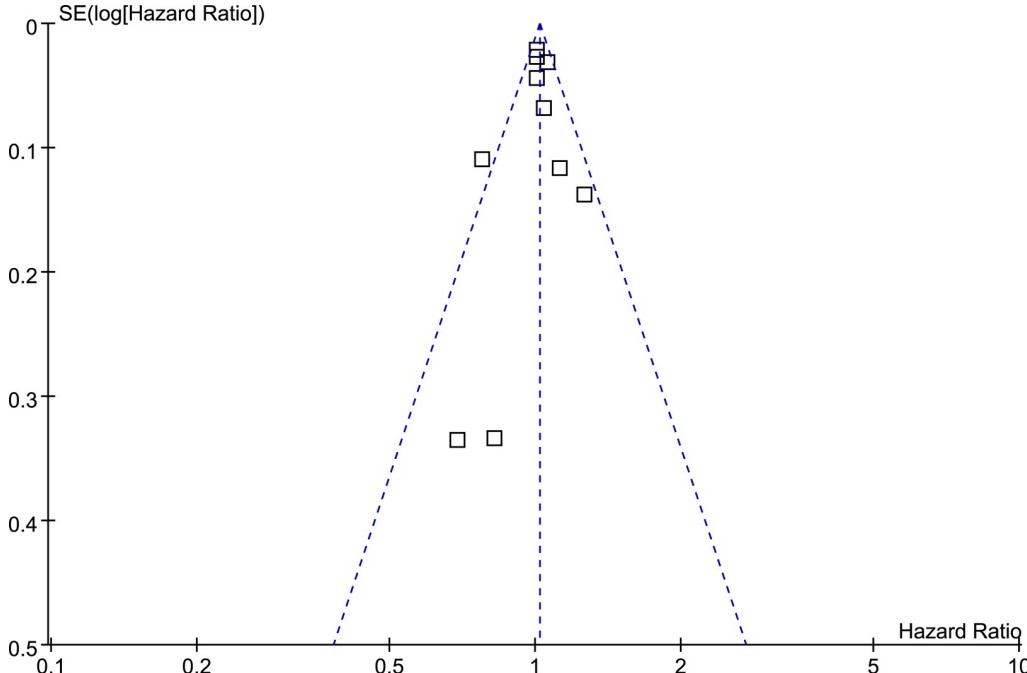

**Fig 9. Funnel plots for the meta-analysis of overall association between BB use and OS of lung cancer.**

with the finding of preclinical studies that propranolol inhibits several pathways involved in breast cancer progression and metastasis. Moreover, preclinical studies have shown that beta-adrenergic agonists may be associated with inhibition of squamous cell growth [35], highlighting the potential diverging association between cancer cell proliferation and some BBs. It has to be mentioned that for most of the included studies, SBB were mainly prescribed ($> 80\%$), and the patients using NSBB were limited. Therefore, the influence of NSBB on survival in lung cancer should better be investigated in large-scale prospective studies or RCTs.

In contrast to the negative finding of this meta-analysis and most of the previous retrospective cohort studies, some recently published small scale studies showed potential benefits of BB on survival in lung cancer patients. In a retrospective study with 77 patients of stage IIIa NSCLC treated with neoadjuvant chemoradiation and subsequent surgery, BB users were associated with a trend of improved OS at 1 year compared with the non-users in univariate analysis (p = 0.08) [39]. Another study with 57 lung cancer patients showed that perioperative intravenous administration of an ultra-short acting SBB landiolol prolonged relapse-free survival in these patients [40], and a phase III RCT has been started accordingly [41]. Results of future prospective studies and RCTs are expected to provide more reliable evidence regarding the association between BB use and OS in patients with lung cancer, preferably according in different histopathological type.

Our study has limitations which should be noticed when interpreting the results. Firstly, only retrospective cohort studies were included. Due to the inherited limitations of retrospective cohort studies, such as recall bias, results of our meta-analysis should be validated in prospective studies. Secondly, patients taking beta blockers prior to their lung cancer diagnosis were likely to have other comorbid conditions that may increase their mortality risks. Typically, BBs are prescribed for patients with cardiovascular disorders, who may also have concurrent medications such as aspirin and statins, which have also been suggested to affect the survival in patients with lung cancer [42]. Although comorbidities and concurrent medications have been generally adjusted among the included studies, these factors may confound the relationship between BB and survival in these patients. Thirdly, since of our study is a meta-analysis based on data of study level, we were unable to provide subgroup data according to smoking history, family history of lung cancer prevalence, or genetic background of individually included patients. We have acknowledged these as potential limitations, which should be considered in designing of future studies. Moreover, although data of adjusted HR were pooled, we could not exclude the chance that there remain residual factors which may confound the association between BB use and OS in lung cancer, such as lung cancer patients with different histopathological type. Future studies are needed for further investigation. In addition, studies included in subgroups for stratified analyses are limited. Therefore, results of stratified analyses should be interpreted with caution. Finally, influences of study characteristics, such as gender and ethnicity of the patients, cancer treatments, comorbidities, and concurrent medications on the outcome could not be evaluated due to these data were rarely reported in the included studies. Further studies are warranted.

## Conclusion

In conclusion, this meta-analysis based on retrospective studies failed to show a significant association between BB use and improved OS in lung cancer patients. Considering the limitations of retrospective studies and generally promising results in preclinical studies, qualified prospective studies and RCTs are needed to investigate the influence of BB use on clinical outcome in lung cancer patients.

## Supporting information

**S1 Checklist.**
(DOC)

## Author Contributions

**Conceptualization:** Zhen Lei, Ying Zuo.

**Data curation:** Zhen Lei, Weiyi Yang, Ying Zuo.

**Formal analysis:** Zhen Lei, Weiyi Yang, Ying Zuo.

**Methodology:** Zhen Lei, Weiyi Yang, Ying Zuo.

**Supervision:** Ying Zuo.

**Validation:** Ying Zuo.

**Writing – original draft:** Zhen Lei.

**Writing – review & editing:** Weiyi Yang, Ying Zuo.

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
