## [Decision Letter · Decision Letter 0]

8 Oct 2020

PONE-D-20-11931

Beta-blocker and survival in patients with lung cancer: a meta-analysis

PLOS ONE

Dear Dr. Zuo,

Thank you for submitting your manuscript to PLOS ONE. After careful consideration, we feel that it has merit but does not fully meet PLOS ONE’s publication criteria as it currently stands. Therefore, we invite you to submit a revised version of the manuscript that addresses the points raised during the review process.

Specifically, the reviewer raised concerns about the reporting of the statistical methodology in the manuscript.

We look forward to receiving your revised manuscript.

Kind regards,

Richard Hodge

Associate Editor

PLOS ONE

Journal Requirements:

2. At this time, we ask that you please provide the full search strategy and search terms for at least one database used as Supplementary Information.

3.Thank you for including the statement that 'The final literature search was performed on January 15, 2020.'

Please revise this statement to clarify whether all databases were searched from inception, or if there were any limits placed on the publication dates in your search."

Reviewers' comments:

Reviewer's Responses to Questions

**Comments to the Author**

1. Is the manuscript technically sound, and do the data support the conclusions?

Reviewer #1: Yes

Reviewer #2: Partly

2. Has the statistical analysis been performed appropriately and rigorously? 

Reviewer #1: Yes

Reviewer #2: Yes

3. Have the authors made all data underlying the findings in their manuscript fully available?

Reviewer #1: Yes

Reviewer #2: Yes

4. Is the manuscript presented in an intelligible fashion and written in standard English?

Reviewer #1: Yes

Reviewer #2: Yes

5. Review Comments to the Author

Reviewer #1: The authors conducts a meta-analysis of 10 studies in order to assess whether there is an association between use of beta blockers and survival in patients with lung cancer. Previous research has hypothesized that use of beta-blockers may be protective for lung cancer patients, but results have been inconsistent. This manuscript aimed to synthesize available studies to provide an updated analysis, including a broader set of studies than prior analyses.

Major Comments:

1. The authors mention multiple other meta analyses looking at this same topic, but there is no discussion of whether the results from this study were consistent with previous meta analyses. Additionally, in the discussion, the authors say that previous meta analyses did not focus on lung cancer. If so, that should be highlighted when discussing how this study is different from prior ones.

2. The authors focused their analysis on survival, however the search terms include outcomes of recurrence, metastasis, progression, surgery, and operation. Please explain why these outcomes were included in the search terms and whether analyses of these outcomes was undertaken.

3. In the section about “Data extracting and quality evaluation”, the authors mention covariates included in the adjustment of RRs. Please clarify whether the outcomes reported were reported as risk ratios or hazard ratios.

4. The methods suggest that included studies needed to have a minimum follow up of one year, however, Table 2 indicates that not all studies had a sufficiently long follow up. Please clarify.

5. Are the authors defining BB use as before or after diagnosis of lung cancer to mean when they started using beta-blockers? Otherwise, it would seem that in figure 4, these are not mutually exclusive categories (ie: patients could have started taking BB before diagnosis, and continued after diagnosis).

6. The results suggest that there may be a benefit to BB in stage III cancers, and those without surgical resection. As surgical resection is not generally considered standard of care for later stage patients (but is for earlier stage patients), the interaction of other therapies besides surgery BB may be an interesting discussion point.

7. Patients taking beta blockers prior to their lung cancer diagnosis likely have other comorb conditions that may put them at higher risk of mortality. This may confound the relationship between beta-blockers and survival in these patients. This should be more clearly defined and discussed.

8. In the discussion of promising studies (page 14-15): why were these studies were not included in the meta analysis?

Minor Comments:

1. The paper could benefit from English language editing

2. Please include a footnote in the tables for abbreviations used

3. It may be beneficial to split figures 3 and 4, as they are almost impossible to read without zooming in.

Reviewer #2: As author concluded that BB failed to show OS in lung cancer patients, I would encourage author to have specific data on patients regarding smoking history, family history of lung cancer prevalence, environmental effect of subgroup of patients and more importantly genetic background such as mutation analysis data which could support the claim in sub groups.

6. PLOS authors have the option to publish the peer review history of their article (what does this mean?). If published, this will include your full peer review and any attached files.

Reviewer #1: No

Reviewer #2: No

---

## [Author Response · Author response to Decision Letter 0]

8 Dec 2020

Response to Reviewer 1

Major Comments:

1. The authors mention multiple other meta analyses looking at this same topic, but there is no discussion of whether the results from this study were consistent with previous meta analyses. Additionally, in the discussion, the authors say that previous meta analyses did not focus on lung cancer. If so, that should be highlighted when discussing how this study is different from prior ones.

Authors’ response: Thanks for your comments. We have compared results of current meta-analysis with those of previous meta-analyses. Moreover, the strengths of current meta-analysis, compared to the previous ones, have also been discussed in the second paragraph of Discussion section of the revised manuscript.

2. The authors focused their analysis on survival, however the search terms include outcomes of recurrence, metastasis, progression, surgery, and operation. Please explain why these outcomes were included in the search terms and whether analyses of these outcomes was undertaken.

Authors’ response: Thanks for your comments. We expanded the search terms regarding the outcomes of the patients to avoid missing of potential related studies. This has been clarified in the revised methods section. Analyses regarding these outcomes were not performed because this meta-analysis focused on the survival outcomes. 

3. In the section about “Data extracting and quality evaluation”, the authors mention covariates included in the adjustment of RRs. Please clarify whether the outcomes reported were reported as risk ratios or hazard ratios.

Authors’ response: Thanks for your comments. It should be “HR”, which has been corrected in the revised manuscript. We apologize for the typo.

4. The methods suggest that included studies needed to have a minimum follow up of one year, however, Table 2 indicates that not all studies had a sufficiently long follow up. Please clarify.

Authors’ response: Thanks for your comments. We have noticed the error in Table 2. We defined adequate follow up as a minimum of one year, and all of the included studies met this criterion. Table 2 has been revised according.

5. Are the authors defining BB use as before or after diagnosis of lung cancer to mean when they started using beta-blockers? Otherwise, it would seem that in figure 4, these are not mutually exclusive categories (ie: patients could have started taking BB before diagnosis, and continued after diagnosis).

Authors’ response: Thanks for your comments. Indeed, we defined BB use as before or after diagnosis of lung cancer according to the time when the patients started using of the medication. This has been clarified in the revised method section.

6. The results suggest that there may be a benefit to BB in stage III cancers, and those without surgical resection. As surgical resection is not generally considered standard of care for later stage patients (but is for earlier stage patients), the interaction of other therapies besides surgery with BB may be an interesting discussion point.

Authors’ response: Thanks for your comments. We have added some discussion regarding these subgroup analyses in the revised manuscript as “Since surgical resection is not generally considered as standard of care for later stage patients with lung cancer, results of these subgroup analyses may reflect the potential adverse interaction of BB with treatment strategies other than surgery in these patients. Indeed, an early study including 107 patients with metastatic NSCLC receiving chemotherapy showed that BB use was associated with improved progression-free survival [1]. In addition, a recent retrospective study also showed that BB use was associated with improved prognosis in NSCLC patients that received immune checkpoint inhibitors [2]”.

7. Patients taking beta blockers prior to their lung cancer diagnosis likely have other comorbid conditions that may put them at higher risk of mortality. This may confound the relationship between beta-blockers and survival in these patients. This should be more clearly defined and discussed.

Authors’ response: Thanks for your comments. We have discussed this in the limitation section of the revised manuscript as “Secondly, patients taking beta blockers prior to their lung cancer diagnosis were likely to have other comorbid conditions that may increase their mortality risks. Typically, BBs are prescribed for patients with cardiovascular disorders, who may also have concurrent medications such as statins, which have also been suggested to favorably affect the survival in patients with lung cancer [3]. Although comorbidities and concurrent medications have been generally adjusted among the included studies, these factors may confound the relationship between BB and survival in these patients”. 

8. In the discussion of promising studies (page 14-15): why were these studies were not included in the meta analysis?

Authors’ response: Thanks for your comments. These studies were not included because they were univariate-analysis studies or did not report HR for overall survival.

Minor Comments:

1. The paper could benefit from English language editing

Authors’ response: Thanks for your comments. Our revised manuscript has been proofread by the Medjaden Bioscience Limited, a professional agency for language editing of scientific research paper, to improve English writing.

2. Please include a footnote in the tables for abbreviations used

Authors’ response: Thanks for your comments. A footnote for abbreviations has been embedded for Table 1, but not for Table 2 because there was not any abbreviations in Table 2.

3. It may be beneficial to split figures 3 and 4, as they are almost impossible to read without zooming in.

Authors’ response: Thanks for your comments. Figure 3 and 4 has been split accordingly as Figure 3-8 in the revised manuscript.

Reviewer #2: 

As author concluded that BB failed to show OS in lung cancer patients, I would encourage author to have specific data on patients regarding smoking history, family history of lung cancer prevalence, environmental effect of subgroup of patients and more importantly genetic background such as mutation analysis data which could support the claim in sub groups.

Authors’ response: Thanks for your comments. In fact, we have performed subgroup analyses according to the adjustment of smoking status in Figure 8 of the revised manuscript, which showed that BB use was not significantly associated with improved OS of lung cancer in studies with and without the adjustment of smoking status of the patients. Since of our study is a meta-analysis based on study level, we were unable to provide subgroup data based on smoking history, family history of lung cancer prevalence, or genetic background of individually included patients. We have acknowledged these as potential limitations, which should be considered in designing of future studies. These contents have been added in the limitation section of the revised manuscript.

References

1. Aydiner A, Ciftci R, Karabulut S, Kilic L. Does beta-blocker therapy improve the survival of patients with metastatic non-small cell lung cancer? Asian Pac J Cancer Prev. 2013;14(10):6109-14. Epub 2013/12/03. doi: 10.7314/apjcp.2013.14.10.6109. PubMed PMID: 24289634.

2. Oh MS, Guzner A, Wainwright DA, Mohindra NA, Chae YK, Behdad A, et al. The Impact of Beta Blockers on Survival Outcomes in Patients With Non-small-cell Lung Cancer Treated With Immune Checkpoint Inhibitors. Clin Lung Cancer. 2020. Epub 2020/09/10. doi: S1525-7304(20)30238-2 [pii]

10.1016/j.cllc.2020.07.016. PubMed PMID: 32900613.

3. Xia DK, Hu ZG, Tian YF, Zeng FJ. Statin use and prognosis of lung cancer: a systematic review and meta-analysis of observational studies and randomized controlled trials. Drug Des Devel Ther. 2019;13:405-22. Epub 2019/02/19. doi: 10.2147/DDDT.S187690

dddt-13-405 [pii]. PubMed PMID: 30774306; PubMed Central PMCID: PMC6350654.

---

## [Decision Letter · Decision Letter 1]

8 Jan 2021

Beta-blocker and survival in patients with lung cancer: a meta-analysis

PONE-D-20-11931R1

Dear Dr. Zuo,

We’re pleased to inform you that your manuscript has been judged scientifically suitable for publication and will be formally accepted for publication once it meets all outstanding technical requirements.

Kind regards,

Jianxin Xue

Academic Editor

PLOS ONE

Additional Editor Comments (optional):

Reviewers' comments:

Reviewer's Responses to Questions

**Comments to the Author**

1. If the authors have adequately addressed your comments raised in a previous round of review and you feel that this manuscript is now acceptable for publication, you may indicate that here to bypass the “Comments to the Author” section, enter your conflict of interest statement in the “Confidential to Editor” section, and submit your "Accept" recommendation.

Reviewer #1: (No Response)

Reviewer #2: All comments have been addressed

2. Is the manuscript technically sound, and do the data support the conclusions?

Reviewer #1: Yes

Reviewer #2: Yes

3. Has the statistical analysis been performed appropriately and rigorously? 

Reviewer #1: Yes

Reviewer #2: Yes

4. Have the authors made all data underlying the findings in their manuscript fully available?

Reviewer #1: Yes

Reviewer #2: Yes

5. Is the manuscript presented in an intelligible fashion and written in standard English?

Reviewer #1: Yes

Reviewer #2: Yes

6. Review Comments to the Author

Reviewer #1: I would like to thank the authors for this revised submission. I have a few additional comments.

1. Table 1 indicates that some studies reported disease free survival, in addition to overall survival. Were these survival outcomes used at all, or was the analysis limited to overall survival. In some places it seems like OS is the only outcome analyzed, but in some places, the authors refer to general “survival outcomes”. If only OS was extracted and analyzed, please specify. If other types of survival were included, please describe how those outcomes were combined.

2. Please define what is meant by selective and non-selective use of beta blockers.

3. Given the large number of subgroup analyses, please address the possibility of false positive/negative results.

4. Where the authors write “Since surgical resection is not generally considered as standard of

care for later stage patients with lung cancer, results of these subgroup analyses may

reflect the potential adverse interaction of BB with treatment strategies other than

surgery in these patients”, please confirm whether adverse interaction is what is meant? The results about those without surgery, along with the next sentence seem to say that research has shown BB to be associated with improved survival in people who receive no surgery/other therapies, not worse survival.

Minor Comments:

1. Figure 2 is listed in the figure legends as the funnel plot, but in the text, it is referred to as Figure 9.

Reviewer #2: Thank you for the response. For future studies please try to underline the patient genetic and environmental factors.

7. PLOS authors have the option to publish the peer review history of their article (what does this mean?). If published, this will include your full peer review and any attached files.

Reviewer #1: No

Reviewer #2: No

---

## [Editor Report · Acceptance letter]

2 Feb 2021

PONE-D-20-11931R1 

Beta-blocker and survival in patients with lung cancer: a meta-analysis 

Dear Dr. Zuo:

I'm pleased to inform you that your manuscript has been deemed suitable for publication in PLOS ONE. Congratulations! Your manuscript is now with our production department. 

Kind regards, 

on behalf of

Dr. Jianxin Xue 

Academic Editor

PLOS ONE